# A neural network model for the evolution of learning in changing environments

**Magdalena Kozielska**[ID]\*, **Franz J. Weissing**[ID]\*

Groningen Institute for Evolutionary Life Sciences, University of Groningen, Groningen, The Netherlands

\* m.a.kozielska@rug.nl; f.j.weissing@rug.nl

## Abstract

Learning from past experience is an important adaptation and theoretical models may help to understand its evolution. Many of the existing models study simple phenotypes and do not consider the mechanisms underlying learning while the more complex neural network models often make biologically unrealistic assumptions and rarely consider evolutionary questions. Here, we present a novel way of modelling learning using small neural networks and a simple, biology-inspired learning algorithm. Learning affects only part of the network, and it is governed by the difference between expectations and reality. We use this model to study the evolution of learning under various environmental conditions and different scenarios for the trade-off between exploration (learning) and exploitation (foraging). Efficient learning readily evolves in our individual-based simulations. However, in line with previous studies, the evolution of learning is less likely in relatively constant environments, where genetic adaptation alone can lead to efficient foraging, or in short-lived organisms that cannot afford to spend much of their lifetime on exploration. Once learning does evolve, the characteristics of the learning strategy (i.e. the duration of the learning period and the learning rate) and the average performance after learning are surprisingly little affected by the frequency and/or magnitude of environmental change. In contrast, an organism's lifespan and the distribution of resources in the environment have a clear effect on the evolved learning strategy: a shorter lifespan or a broader resource distribution lead to fewer learning episodes and larger learning rates. Interestingly, a longer learning period does not always lead to better performance, indicating that the evolved neural networks differ in the effectiveness of learning. Overall, however, we show that a biologically inspired, yet relatively simple, learning mechanism can evolve to lead to an efficient adaptation in a changing environment.

**Data Availability Statement:** The source code (in C + +) for the simulation program, example output files and summary data are available at https://doi.org/10.34894/IYWJZM.

## Author summary

The ability to learn from experience is an important adaptation. However, it is still unclear how learning is shaped by natural selection. Here, we present a novel way of modelling the evolution of learning using small neural networks and a simple, biology-inspired learning mechanism. Computer simulations reveal that efficient learning readily evolves in this model. However, the evolution of learning is less likely in relatively constant environments (where evolved inborn preferences can guide animal behaviour) and in short-

**Funding:** This work was supported by the European Research Council (ERC Advanced Grant No. 789240 awarded to F.J.W.). The funders had no role in study design, data collection and analysis, decision to publish, or preparation of the manuscript.

**Competing interests:** The authors have declared that no competing interests exist.

lived organisms (that cannot afford to spend much of their lifetime on learning). If learning does evolve, the evolved learning strategy is strongly affected by the lifespan and environmental richness but surprisingly little by the rate and degree of environmental change. In summary, we show that a simple and biologically plausible mechanism can help understand the evolution of learning and the structure of the evolved learning strategies.

## Introduction

Learning can be defined as a change in the nervous system manifested as altered behaviour due to experience [1]. The ability to learn is widespread in the animal kingdom as it is an important adaptation to life in complex environments [2–5]. Learning has been studied extensively in different fields, like psychology [6], ethology [7], neurobiology [8], and more recently artificial intelligence [9]. A number of theoretical studies have been conducted in order to understand the evolution of learning (see [7,10,11] and references therein) but many questions still remain unanswered [7].

The limited progress may be related to the fact that only a few evolution-oriented theoretical studies considered that experience-based changes in behaviour are achieved via changes in neural networks. Many studies take a behavioural gambit approach [12], assuming that mechanisms do not matter and that evolution will always shape learning in such a way that the outcome is optimal. Modelling studies that do take mechanisms into consideration, typically focus on the evolution of simple learning rules that are determined by a small number of parameters (e.g. [10,11,13–16] but see [17]). It is difficult to imagine how such rules could emerge via the evolution of brain plasticity.

In contrast, machine learning and artificial intelligence deal with complex neural networks capable of learning. However, the proposed learning mechanisms often rely on complicated algorithms, such as backpropagation, which affect all connections in the network [18,19]. Such techniques are very efficient in machine learning applications but are far removed from biological reality [20]. Additionally, when evolutionary considerations are included, they are usually limited to network optimisation instead of asking evolutionary questions (but see [21,22]). Mutation and selection are usually viewed as useful tools that can be freely designed to achieve a computationally efficient outcome and often unrealistic assumptions are made about the way how natural selection works [23–25].

Here, we take a first step toward filling the gap between these two approaches. We study the evolution of neural networks that are capable of learning via a simple but plausible mechanism. In our model, neural networks can change over the generations through evolution by natural selection, but also through learning within the lifetime of an individual. Experience-induced changes in the network are localized and affect only a small number of neural connections–an approach inspired by "reservoir computing" [26,27]. The learning mechanism is based on prediction error, the difference between an animal's expectation and observed reality. Such a mechanism is biologically plausible, as prediction errors are signalled by the neurotransmitter dopamine, which is ascribed an important role in learning [28,29]. Learning algorithms based on error prediction have also been successfully implemented in machine learning applications [26]. To our knowledge, this is the first time that such a biology-inspired local learning mechanism is implemented in a study on the evolution of learning in biological organisms (see [22,30] for analogous models in robotics). Yet, we would like to stress that it is not our goal to build a realistic model of a brain. Rather, we view our model as a conceptual tool to explore how the addition of mechanistic detail affects the evolution of learning.

Learning theory predicts that the degree of environmental change affects the adaptive value of learning and therefore the probability that learning evolves [7]. Generally speaking, learning is expected to be most advantageous for moderate rates of environmental change [31]. If there is little or no change, genetic control of behaviour should evolve. Learning is also not profitable if the change is too frequent because information on the environment gets outdated too fast. We therefore study the evolution of neural networks and their learning mechanisms in different regimes of frequency and magnitude of environmental change, as it allows us to investigate whether a more mechanistic implementation of learning is in line with the predictions of "mechanism-free" theory.

We also touch upon a rarely studied aspect of learning theory–the effect of lifespan on the evolution of learning. As far as we know there are only two studies addressing this question [32,33]. These models are vastly different from each other in assumptions and ecological context and lead to different predictions on whether the investment in learning should be highest for shorter or longer lifespans. To add to this limited body of knowledge, we also study how different lifespans affect the evolution of learning and the evolved learning strategy.

Our simulation study addresses the following research questions: Under which environmental conditions does learning evolve? What is the evolved learning strategy and how efficient is it? What is the effect of lifespan on the evolved learning strategy?

## Methods

### Model overview

We use individual-based simulations to study the evolution of simple neural networks that are able to adapt to changing environment by learning. In our model, individuals harbour a neural network that guides their foraging decisions. Individuals have a fixed lifetime of a given number of timesteps. At the start of their life, they can spend a number of timesteps on learning. During this learning period, they gather information about the quality (energy content) of food items in their environment, and they use this information to adjust their neural network. After the learning period, individuals switch to foraging, when they use their network to assess the available foraging options. They choose the food item their neural network finds the most profitable, consume it and gain energy equal to its quality. The more energy their gather during the whole foraging phase, the more offspring they have. There is a trade-off between exploration and exploitation: the longer the learning ('exploration') period, the shorter the foraging ('exploitation') period, but potentially the higher the efficacy with which the foraging period is used.

Offspring inherit their neural network and their learning strategy (the duration of the learning period and the learning rate) from their parents, subject to rare mutations. Environmental conditions can change between generations, making learning a potentially adaptive strategy.

### Neural networks

Each individual possesses a neural network that is used to predict the quality of food items on the basis of environment-specific cues. We consider relatively simple neural networks consisting of 10 neuron-like nodes (Fig 1). In a pilot study, we also considered more complex networks, but the network considered here performed almost as well as more complex networks while being faster. Our network receives a cue $C$ as input and it produces an output $Q_p(C)$ that can be interpreted as the predicted quality of a food item emitting that cue. The networks consist of nodes (the circles in Fig 1) that are organized in a sequence of layers. Each node is connected to one or several nodes in the subsequent layer (the arrows in Fig 1), and it can stimulate or inhibit the activities of these nodes.

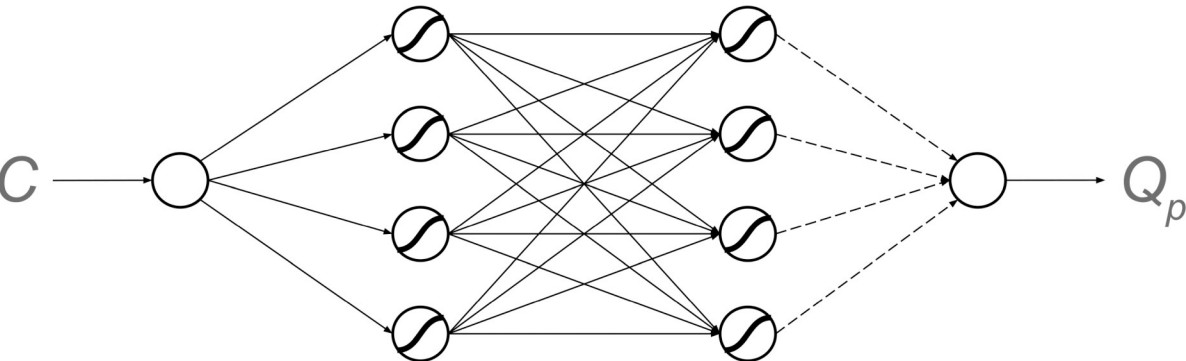

**Fig 1. The neural network used in this study.** Our network receives a cue $C$ as input and it produces an output $Q_p$ that is the predicted quality of a food item emitting that cue. In this model we use a network with one input and one output ($C$ and $Q_p$, respectively) and two hidden layers, each with four nodes. Arrows indicate the information flow in the network. Solid arrows represent genetically hardwired connection that do not change during learning. Dashed arrows represent the weights that are genetically determined but can also change during learning (see text for more details).

Each connection has a certain strength—weight $w$, where a positive value of $w$ represents stimulation, while a negative value corresponds to inhibition. The input node receives the cue value $C$ which is a real number. This value is processed and determines the node activities at the subsequent level. More precisely, the activity $y_i$ of node $i$ in each layer is given by an expression of the form

$$y_i = A\left(\sum_j w_{ij} x_j + b_i\right) \tag{1}$$

Here $j$ runs over all nodes of the previous layer that are connected to $i$, $x_j$ is the activity of node $j$, and $w_{ij}$ is the strength of the connection between nodes $j$ and $i$. $b_i$ is the baseline activation of node $i$. Function $A$ is a so-called "activation function." Such functions can be useful, as they allow for more versatile input-output relationships of neural networks and because they can ensure that the activity levels $y_i$ are restricted to a certain range (such as the interval [0,1]) [34]. A preliminary test showed that the results are not strongly affected by the applied activation function. In this paper, we used the"clamped ReLU" function that is fast and returns 0 if given values lower than 0, and 1 for the values larger than 1. For values between 0 and 1, it returns these values without transformation. No activation function was used for the output node.

We assumed that the network architecture does not change throughout the simulation and that all the network parameters $w_{ij}$ and $b_i$ are heritable and transmitted from parents to offspring (subject to mutation, see below). Therefore, the strength of connections between the nodes and the baseline node activations can change in the process of evolution. Additionally, some weights can change during the individual's lifetime via learning.

## Learning

Four weights of the network connected to the output node (dashed arrows in Fig 1) can be modified via learning during the individual's lifetime.

Different methods for network learning are used in artificial intelligence applications. Many of them implement relatively complex algorithms that change all (or most) weights of the network based on global information (e.g. error backpropagation) and are therefore unrealistic from the biological point of view. As far as we know, there is no experimental data supporting such learning happening in the brain. Additionally, at the initial stages of the evolution of learning a simple learning algorithm is more likely to evolve from scratch.

Therefore, we decided to use a simple learning method, inspired by reservoir computing [26,27], that assumes that learning is more localised in the brain and that it only leads to changes in weights of the last layer, i.e. weights that directly influence the output of the network. These changes are governed by the so-called "Delta Rule" that has proven to be effective in reservoir computing and other machine learning applications [26]. It uses the difference between the current network output (prediction) and the feedback received from the environment as a teaching signal. Interestingly, local dopamine concentrations in the brain may signal such prediction error [28,29,35] and prediction-error-based learning is a well-known phenomenon in animal psychology research [6].

Therefore, the changes in weights of the last layer (connected to the output node; dashed arrows in Fig 1) after one round of learning are given by the 'Delta Rule' [34]:

$$\Delta w_{ij} = L(Q - Q_p)x_j \tag{2}$$

where $\Delta w_{ij}$ is the change in the weight connecting node $j$ in the preceding layer to the output node $i$; $Q$ is the actual quality of the food item; $Q_p$ is the quality predicted by the network before the weights are updated; $x_j$ is the activation level of node $j$; and $L$ is the learning rate, a heritable factor that determines how strongly the network weights are modified during each learning event. For a given prediction error, a higher learning rate leads to larger change in the four network weights affected by learning.

It should be noted that the values of the modified weights are not passed to the offspring, but only the genes specifying the weights' values at the beginning of life.

## Environmental change

Individuals live in an environment that contains food items of different quality. Food items can be distinguished on the basis of their properties (like colour or smell) that we will call 'cues'. For simplicity, we assume that the cues can be arranged on a circle or, equivalently, on the 'wrapped' interval $[-1,1]$, where -1 corresponds to +1. The energetic quality $Q$ of a food item emitting cue $C$ is given by a Gaussian function (see Fig 2):

$$Q(C) = \exp\left( -\frac{1}{2} \cdot \left( \frac{C - P}{\sigma} \right)^2 \right) \tag{3}$$

where $P$ is the location of the peak of the Gaussian while $\sigma$ describes the width of the function. In our simulations, the peak was initially located at zero. As a default, we used $\sigma = 0.25$, but other values were also tested.

We assume that the environment can change over the generations, in the sense that the quality associated with specific cues may change. We model this by shifting the whole quality function randomly to the left or to the right. As mentioned above, we wrapped the quality function in order to assure that the total amount of resources in every generation is the same (see Fig 2).

In order to study the effect of environmental change on the evolution of learning, we considered (a) different frequencies of environmental change ($f$)–we present results for the values $f = \{1, 0.1, 0.01\}$, corresponding to a change every 1, 10 and 100 generations, respectively; and (b) different magnitudes of environmental change ($m$)–how much the peak of the quality distribution shifts (left or right) when the environment changes. Throughout the simulations we used values $m = \{0.1, 0.25, 0.4\}$. Each time the peak moved, a small error term (with a coefficient of variation of 5%) was added to $m$ to prevent that only finitely many peak locations would be experienced in the course of evolution.

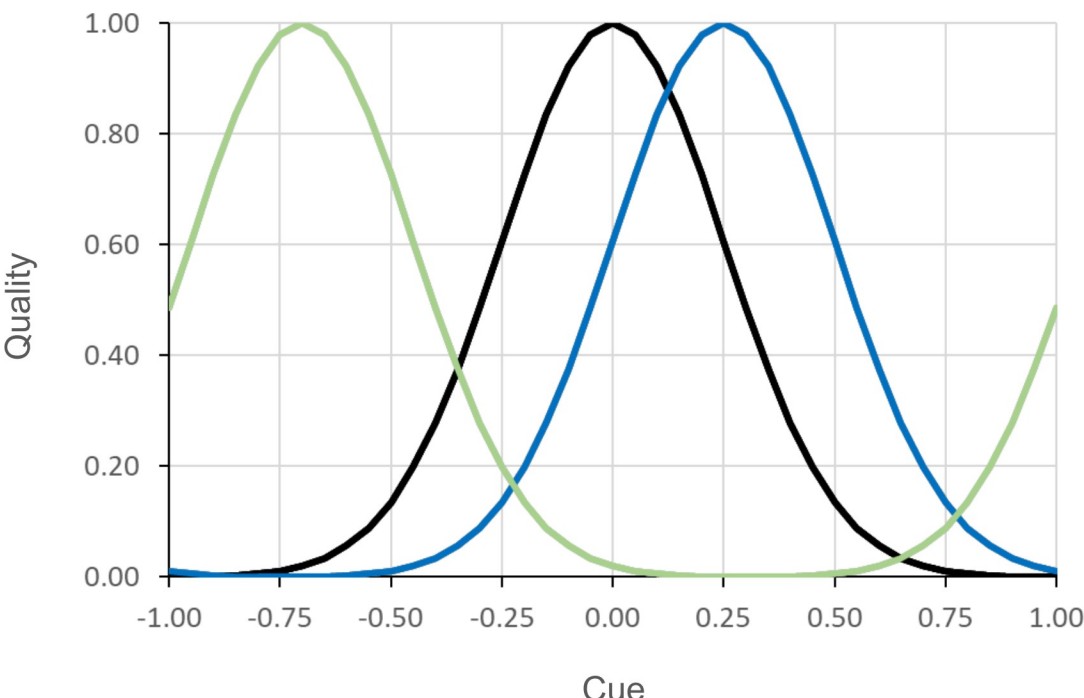

**Fig 2. Relationship between food cues and food quality at three different points in time.** Three Gaussian functions with σ = 0.25 illustrate the "environmental profile" (Eq 3) at three time points. The environment changes via a shift of the peak *P* of the profile.

### Life history

The lifespan of each individual is divided into a fixed number of discrete timesteps. We focus on a lifespan of 500 timesteps but later also briefly discuss the effect of shorter lifespan on the outcome of evolution. The first part of life is spent on learning and the duration of the learning period is either a fixed parameter or a heritable property. The second part of life is spent on foraging, where previous learning can potentially improve the ability to choose food items of better quality. At the end of their life, individuals reproduce and then die.

The implicit assumption that all learning happens at the beginning of life is reasonable in view of our assumption that the environment is constant within a generation and only changes between generations. In such a scenario, learning should take place as early as possible, in order to profit maximally from it during foraging. Of course, the environment may also change within generations, necessitating life-long learning. Our model could accommodate for this (see Discussion), but we leave the analysis of this more complicated scenario to a future attempt.

### Learning period

In each timestep of the learning period, each individual explores one food item: it gets one randomly chosen cue, predicts the food quality associated with that cue (using its current neural network), is informed about the true food quality of the corresponding food item, and updates its network accordingly (see above). Once the learning period is finished, the network resulting from the succession of learning steps is used to guide the individual's decisions in the foraging phase. To reduce one source of randomness from our simulations, all individuals in the population were presented with the same sequence of random cues during the learning period.

### Foraging period

In each timestep of the foraging period, each individual is presented with 5 food items. Based on the food properties (cues) an individual has to decide which item to consume. To this end, the individual uses its neural network (that is partly inherited and partly adjusted by learning) to predict the quality of the food items presented. Subsequently, it consumes the item it predicted to be the best, gaining energy equal to the food item's true quality. In the next timestep, a new set of food items is presented, etc. Again, we reduced the randomness by presenting the same sets of food items to all individuals in the population.

### Reproduction and inheritance

In our simulations, we consider a population of 1000 haploid individuals and discrete, non-overlapping generations. Each individual harbours genes that encode the (initial) connection weights and biases of its neural network (in total 33 values), the learning rate and the number of learning episodes. Weights and biases can take any real value, the learning rate can be any non-negative real number and the length of the learning period is an integer ranging from zero to the lifespan of the organism.

Individuals reproduce after the foraging period. The expected reproductive success of an individual is proportional to the total amount of energy the individual gathered during its entire foraging period. Offspring are produced via a lottery method [36]: For each offspring, a parent is drawn at random (with replacement); the probability that a given individual is drawn as a parent is proportional to the individual's total energy gained. The offspring inherits all network parameters, learning rate and the number of learning episodes from its parent. Per locus, the parental allele mutates with probability 0.01. When a mutation occurs, a small number (the mutational step size) is added to the parental value. For weights, biases and learning rate the mutational step size is drawn from a normal distribution with mean 0 and standard deviation 0.1. For the locus encoding the number of learning episodes (which is a non-negative whole number), a mutation either leads to the increase or to the decrease by one unit, both with equal probability. When all offspring are produced, the offspring population replaces the parental population and the new generation starts.

### Simulation setup

Each simulation started with a population consisting of individuals with random parameter values. Initial weights and biases were drawn from a uniform distribution U(-1,1) and the learning rate from the uniform distribution U(0,1). The number of initial learning episodes was set to a specific value depending on the type of simulation (see Results).

Most simulations were run for 50,000 generations, but evolutionary equilibrium (judged by the population average of the amount of gathered energy) was usually reached in a much shorter time. All parameters investigated in this study are listed in Table 1.

## Results

We present our results in three sections. In the first section, the duration of the learning period is fixed. This removes the exploration-exploitation trade-off and allows us to focus on the evolution of the network and the learning mechanism. In the second section, we investigate the joint evolution of the network and the duration of the learning period in order to study the trade-off between exploration and exploitation. In the first two sections, we focus on organisms with a lifespan of 500 time units and an intermediate width of the environmental quality

**Table 1. Model parameters and the values investigated in this study.** Default values of the parameters are marked in bold.

| Symbol | Meaning | Values |
|---|---|---|
| LE | Number of learning episodes | 0, 5, 10, 20, 40* Evolving** |
| $L$ | Learning rate (influences how much the network changes during each learning event; see Eq (2)) | Evolving |
|  | Lifespan | 50, **500** |
| $f$ | Frequency of environmental change (environment remains constant for $1/f$ generations) | 0.01, 0.1, 1 |
| $m$ | Magnitude of environmental change–average change of the environmental peak location | 0,1, 0.25, 0.4 |
| σ | Width of the environmental profile | 0.1, **0.25**, 0.4 |

* In the first section of the Results, LE was set to a fixed value without possibility to mutate.

** When LE was allowed to evolve, it was initially set to 20.

distribution (σ = 0.25). In the last section, we study how the evolutionary outcome is affected by changes to these parameters.

## Fixed duration of the learning period

As a first step, we fixed the number of learning episodes (LE) to different values. For simplicity, we present the results for a lifespan of 500, but the same pattern is seen for other lifespans, as long as the absolute number of learning episodes is the same.

One of the most important measures for the performance of a network is its ability to choose a high-quality food item among the available options. We calculated the relative performance ("performance" in brief) of a network within a given choice situation by dividing the energy content of the food item chosen by the energy content of the highest-quality food item on offer. Therefore, performance equals one if during the foraging period an individual always chooses optimally, and it is smaller than one otherwise. All other things being equal, a higher performance leads to higher fitness, which in our model is proportional to the "lifetime energy gain", that is, to the sum of the energy of all food items collected throughout the lifetime. When, however, the learning periods differ, a higher performance does not necessarily result in a higher lifetime energy gain. If the higher performance is associated with a longer learning period and, hence, a shorter foraging time it may not offset the time "lost".

Fig 3 shows how, for a given duration of the learning period, the performance and the lifetime energy gain of the evolved learning networks are affected by the frequency and magnitude of environmental change. As a benchmark, consider first the absence of learning (LE = 0; red dots and lines). For the environmental quality distribution considered here (σ = 0.25), a randomly choosing individual would achieve a performance of about 0.41. Even without learning, the networks typically perform better than this, because they adapt genetically to the pattern of environmental change. Such "adaptive tracking" [37] can lead to a high network performance if the environmental change is rare and/or if the magnitude of environmental change is small. In the case of $f$ = 0.01 (change once every 100 generations) and $m$ = 0.1 (small magnitude of change), the non-learning networks achieve practically the same high performance as the networks that were allowed to learn. In this case, the non-learning networks even have a fitness advantage (a higher lifetime energy gain), as they do not lose foraging time. On the other hand, if $f$ = 1 (change every generation) and $m$ = 0.4 (large magnitude of change), the genetic mechanism cannot adaptively track the environmental changes, and the networks do not perform better than 0.41, the performance of a random-choice mechanism.

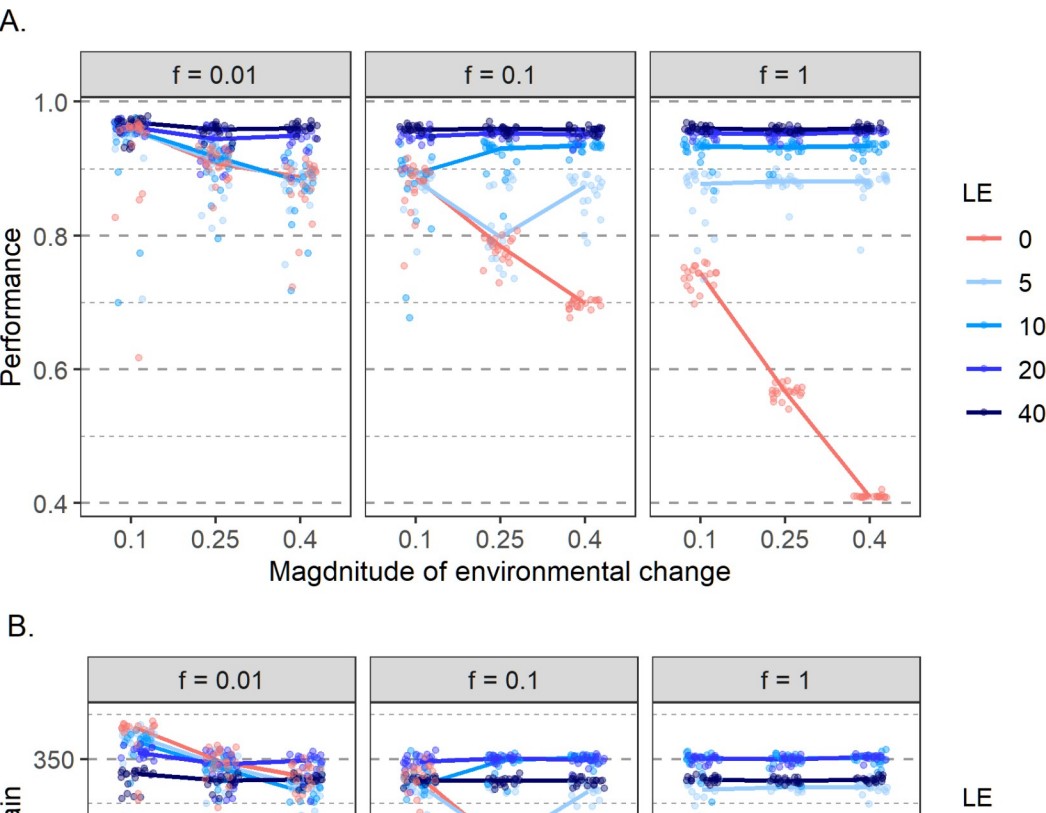

**Fig 3. Effect of a fixed number of learning episodes (LE) on (A) evolved network performance and (B) lifetime energy gain in different environmental regimes.** Panels in different columns corresponds to a different frequency of environmental change *f*, ranging from 0.01 (a change once every 100 generations) to 1.0 (a change every generation). The *x*-axis of each panel represents the magnitude of environmental change: the distance that the environmental quality peak moves when change occurs. 20 replicate simulations were run for each parameter combination (in all cases, lifespan = 500). Each replicate is represented by a coloured point, which corresponds to the population mean of this replicate, averaged over the last 2000 generations. The lines connect the median values of 20 replicates for different parameter settings. As expected, performance tends to increase with the number of learning episodes. However, the total amount of resources gained tends to be highest for an intermediate number of learning episodes, because a longer learning period reduces the time left for foraging.

As expected, when learning is present, a longer learning period has a positive effect on network performance (Fig 3A). In general, performance increases with the number of learning episodes, but levels off from a certain point onward. In other words, the returns from adding more learning episodes diminish and eventually they become negligible. Therefore, the foraging time lost to longer learning can to some extent be compensated by improved performance,

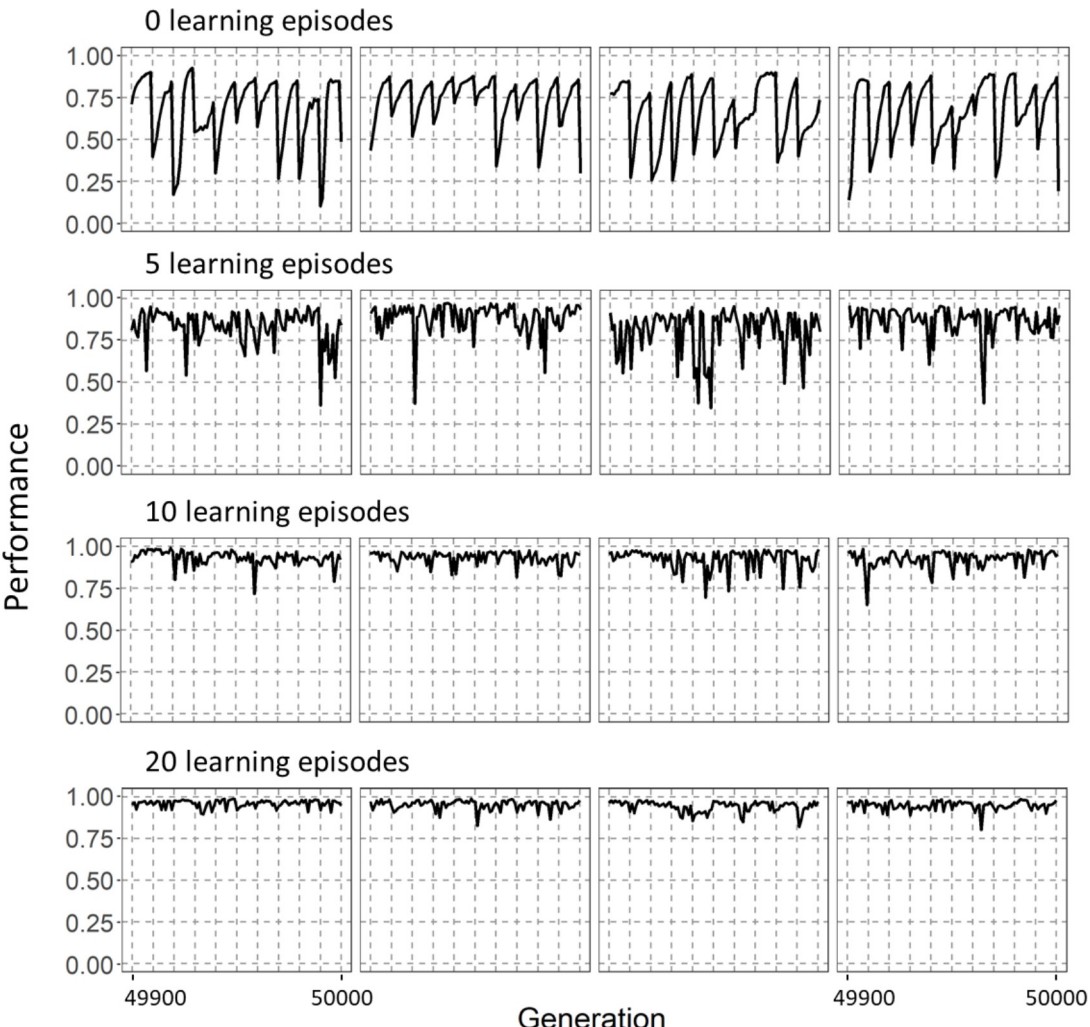

**Fig 4. The time course of network performance in a changing environment.** The panels show the time course of average population performance over the last 100 generations of simulations with an environmental change rate $f$ = 0.1 (change once every 10 generations) and magnitude $m$ = 0.4. For four values of the number of learning episodes (LE = 0, 5, 10, 20) four randomly chosen replicate simulations are shown. In the absence of learning (LE = 0), the population performance clearly drops to low levels every time the environment changes (indicated by vertical dashed lines). With an increasing number of learning episodes, the drops in performance are smaller, and performance is better throughout the simulation.

but there is a limit to that. Accordingly, fitness (lifetime energy gain) is typically maximized for an intermediate number of learning periods (Fig 3B).

Fig 4 shows the temporal dynamics in more detail, for a scenario where the environment changes every ten generations. Without learning (LE = 0), performance drops every time when environmental change happens and subsequently recovers (due to the genetic evolution of the network) to its former value. Learning reduces the drop in performance considerably, especially if the learning period is long (LE = 20).

Quite generally, learning strongly improves the match between the real environmental quality and the one predicted by individuals if the environment changes frequently and/or if the magnitude of change is large. This is exemplified in Fig 5 which shows prediction profiles for networks evolved for different durations of the learning period. When there is no learning and the environment changed, the prediction profile does not match the real quality associated

with different cues. With learning, the innate quality prediction can be relatively poor, as it is considerably improved by learning. Even though the predicted quality after learning does not perfectly match the real environmental quality for all cues, individuals perform quite well, as they only need to assess the relative quality associated with the five cues available during one foraging episode and to find the one that is linked with the highest energy level.

Conversely, if environmental change happens less frequently or is smaller in magnitude, even though individuals spent time learning they are not necessarily efficient at it and a considerable number of learning episodes is needed to reach an improvement in performance (Fig 3A in the main text and Fig A in S1 Appendix).

Examples of the evolved neural networks and the effect of learning on the weights can be seen in Fig 6. In different replicates, different networks evolved, but within a replicate variability between individuals was low.

We also looked at the evolved learning rate. In environmental settings in which learning increases performance (i.e., when environmental change is frequent and/or of large magnitude), the learning rate tends to slightly decrease with the increasing learning period (Fig B in S1 Appendix). In other words, when the learning period is brief, learning tends to take place in

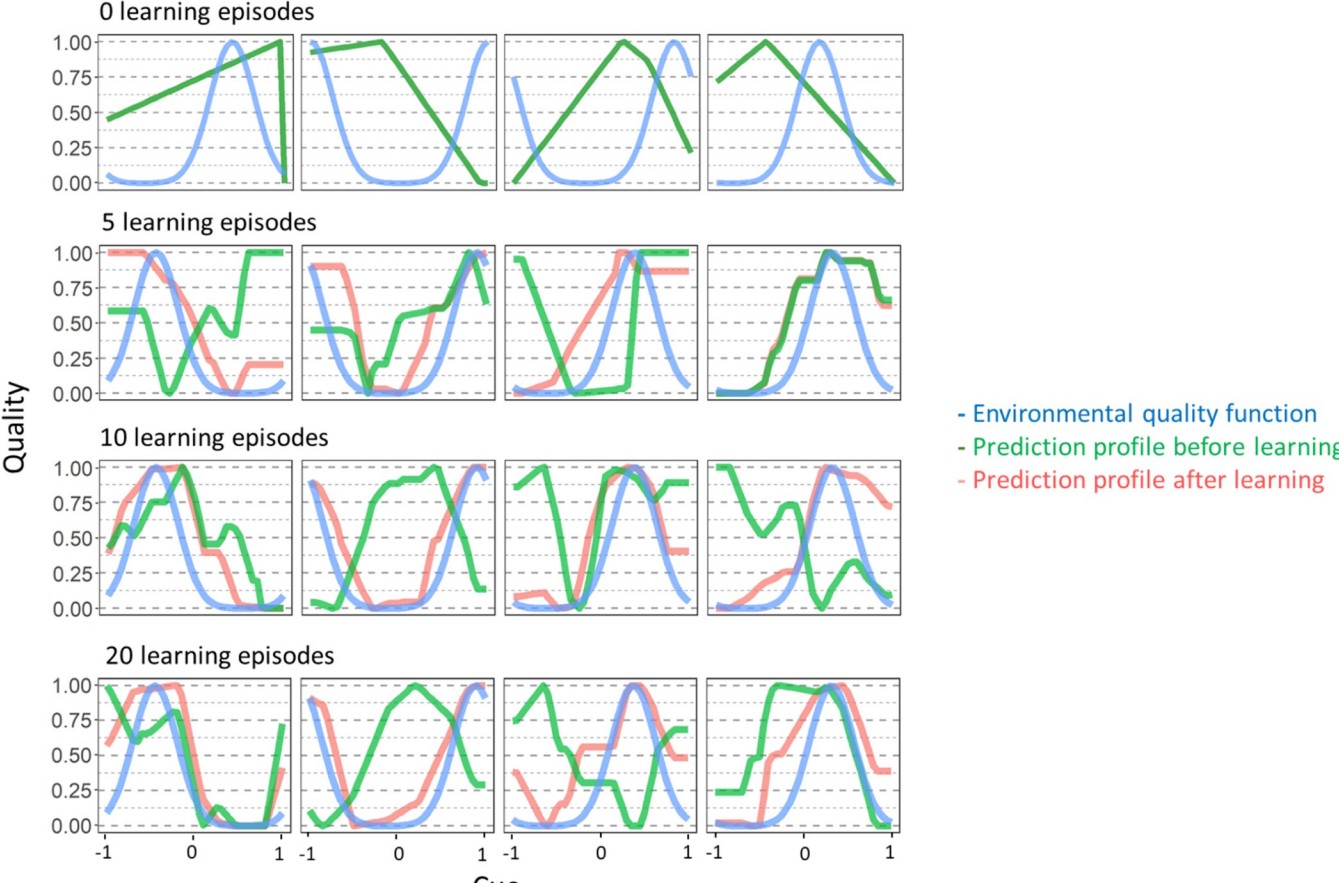

**Fig 5. Prediction profiles of networks evolved for different durations of the learning period.** For each of the 16 populations presented in Fig 4, one network was chosen at random at the end of the simulation (when the environment had just changed) and was investigated in more detail. The plots show the environmental quality function (blue), the prediction profile (i.e., the quality predicted for each possible cue) of the network before learning had started (green) and at the end of the learning period (red). For longer learning periods, the "learned" prediction profiles (red curve) match the "true" environment quality profile (blue curve) reasonably well, even though the "innate" prediction profile (green curve) is way off target. Parameter settings as in Fig 4 ($f$ = 0.1, $m$ = 0.4).

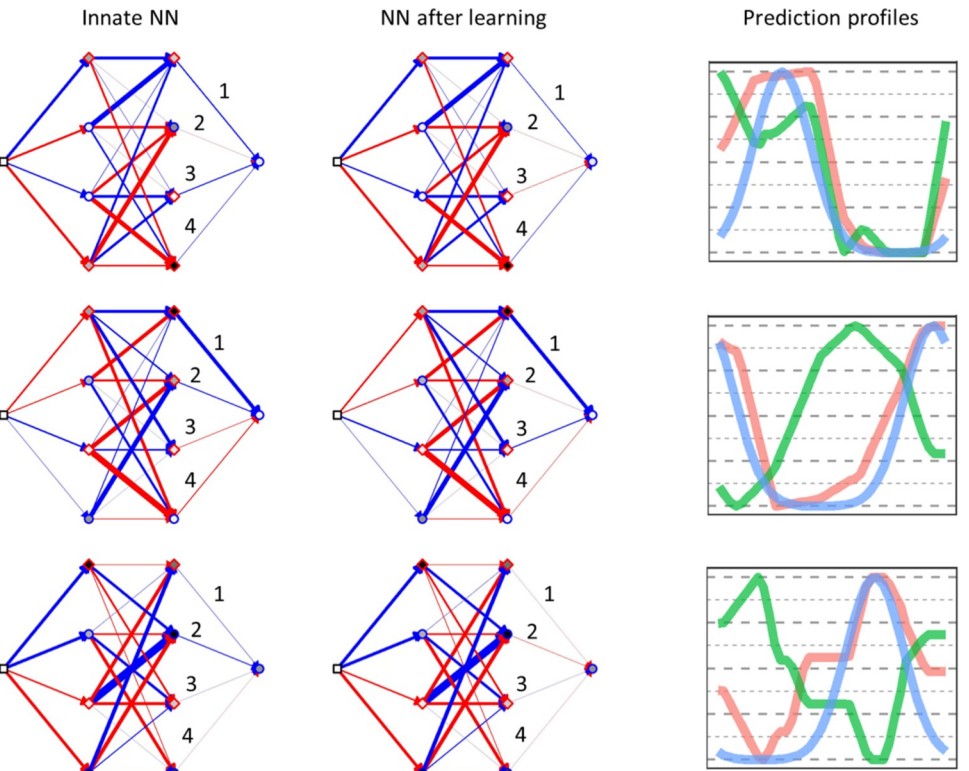

**Fig 6. Examples of evolved neural networks and their prediction profiles before and after learning.** Networks from three different replicates from simulations with LE = 20 are shown (three of the four individuals shown in Fig 5). Blue arrows correspond to excitatory connections (positive weights) and red to inhibitory connections (negative weights) The thickness of the lines is proportional to the strength of the connection. The baseline activation of each node is represented by circles with a blue edge for positive values and by diamonds with a red edge for negative values. The absolute strength of the baseline activation is given by the inner shading of the symbol, the darker the colour the larger the value. During learning only the four numbered weights can change. For example, for the network in the centre, two weights changed the strength and the type of connection (from excitatory to inhibitory—weight 2; and the other way round–weight 3) and weight 4 weakened in strength. Such relatively small changes to the network lead to a drastic change in the prediction profile (right column, plotting convention as in Fig 5). Note that the four "learning weights" of the networks tend to have smaller absolute weight values (thinner lines) than other weights. This was a common pattern for networks that evolved efficient learning (Fig C in S1 Appendix).

larger strides, while it tends to take place in smaller steps when there is much time for learning.

## Evolution of the duration of the learning period

From now on, we assume that the duration of the learning period coevolves with the learning rate and the properties of the network. Accordingly, the evolutionary outcome reflects the exploration-exploitation trade-off: the system should evolve to a state in which the time spent learning (and lost for foraging) is compensated by the better choices made during foraging. There is, however, a start-up problem. When a simulation starts with a random network, the learning period may evolve toward zero (as learning does not provide any benefit at the early stage of network evolution). Once the learning period is around zero, and the selection for learning is weak, the evolution of the optimal number of learning episodes is hampered. To overcome this problem, we started each simulation with a fixed number of learning episodes (LE = 20) and let the network and the learning rate evolve for 10K generations. After this initial

period, the number of learning episodes became subject to mutations and could therefore start to evolve as well for further 50K generations.

Fig 7 shows that learning persists in the population under a broad range of environmental conditions, leading to greatly increased performance and fitness (compared to a population with no learning; Fig 7A). This shows again that the time spent learning can be compensated by the better choices made in the foraging phase. In other words, the benefits of learning often outweigh the costs in terms of a shortened foraging time.

Whether learning evolves can be to a large extent predicted from the results of the previous section. Fig 3 shows what number of learning episodes (among the ones tested) lead to the highest gain in resources, and, hence fitness. For example, when the environment changes every generation the simulations with learning episodes of 10 and 20 obtained the highest energy. When we let the number of learning episodes evolve it reaches a median value of around 17 (Fig 7B). On the other hand, when a small change of 0.1 happens every 100 generations, then the highest energy gain was obtained in simulations with no learning. In line with this, the number of learning episodes converges to zero, corresponding to the loss of learning from the population.

Smaller and less frequent environmental changes reduce the likelihood that learning is maintained. Interestingly, when learning evolves, the number of learning episodes, network performance, and network fitness are practically independent of the magnitude or frequency of environmental change (Fig 7). This seems to be due to the fact that when effective learning evolves, the networks do not track environmental change genetically but fully rely on learning. In other words, the "innate prediction profile" is relatively stable over time, while the "learned prediction profile" is adjusted to the current environmental quality function (see Fig D in S1 Appendix).

In addition to the duration of the learning period (the number of learning episodes), the learning rate is also an important aspect of the learning strategy. This parameter is also practically independent of the environmental conditions in the replicates in which the efficient learning evolved (Fig 7C).

Fig 8 illustrates that the relationship between the number of learning episodes and performance is not straightforward. First of all, in some environmental conditions, the outcome of evolution differs considerably across replicates. If that is the case, a lower number of learning episodes tends to be linked with lower performance (Fig 8). This often results in lower fitness, i.e., the additional time spent foraging does not compensate for the lower performance. However, similarly to what we observed in the previous section, when learning evolves, a longer learning period does not always lead to better performance. In some environmental regimes, a wide range of learning episodes leads to practically the same performance (Fig 8). We hypothesize that the variability between replicates might be partly explained by the properties of the coevolved networks. Each replicate evolves a unique network that likely affects the effectiveness of learning. Loss of learning in some replicates even if retention of learning could potentially lead to higher fitness could be linked to the learning ability of a specific network. However, due to the complexity of the system, we were not able to prove this hypothesis. It is also worth noting that in environmental regimes that lead to either loss or maintenance of learning, in some replicates in which learning was lost at some point, it revolved again.

Within each environmental regime, a range of learning episodes evolve. One might expect that in replicates in which the learning period is shorter, the learning rate is higher to allow for faster adjustment of weights and *vice versa* (similarly to the situation with fixed LE). When efficient learning evolves, in most environmental regimes there is indeed a negative relation between the learning rates and the number of learning episodes, but this trend is weak and a lot of variation is present (Fig 9). If the number of learning episodes is very small (below 5),

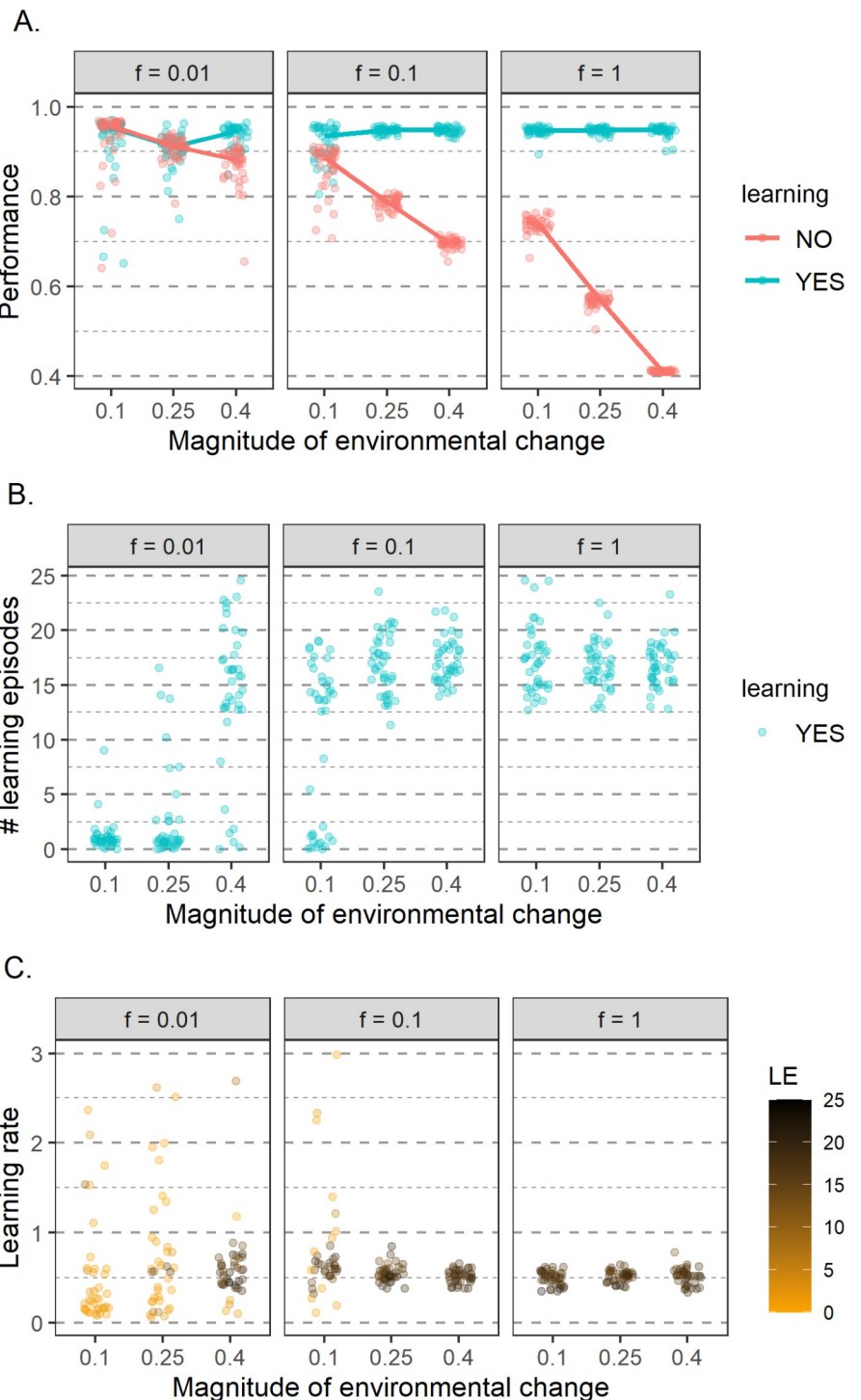

**Fig 7. Joint evolution of (A) network performance, (B) duration of the learning period, and (C) learning rate for various environmental scenarios.** Parameter settings and graphical conventions are as in Fig 3. In (A), the performance of the evolved networks in the simulations in which learning was allowed to evolve is shown in turquoise. For comparison, the simulations in Fig 3A where learning was not allowed to evolve (LE = 0) are also shown (in red). (B) shows the evolved number of learning episodes. Notice that LE often evolves toward zero (i.e., learning disappears in the course of evolution) when the magnitude of change is small and environmental change is infrequent. (C) shows the evolved learning rates–different colours indicate the association between the evolved learning rate and the evolved duration of the learning period in the replicate simulations. Notice that the evolved learning rate is close to 0.5 in all simulations where learning evolved (LE > 10). When learning disappeared in the course of evolution (LE < 5), the

learning rate is no longer under strong selection and can take on many different values (eight data points with LE < 5 are not visible, as the learning rate exceeds 3). The points are semi-transparent and darker spots indicate that multiple replicates evolved the same value.

learning is not effective (Fig 8) and the learning rate is very variable between replicates, consistent with random drift (Fig 9).

The last, but crucial part of the evolved learning mechanism is the neural network itself, and specifically the value of its weights. Neural networks are notoriously difficult to analyse but we decided to look at some of the properties of evolved neural networks. The only pattern we could see is that the average strength of connections that can be adjusted through learning was clearly lower for networks that evolved learning compared to networks that did not evolve learning (Fig C in S1 Appendix). This suggests that efficient learning cannot be achieved when weights are too high.

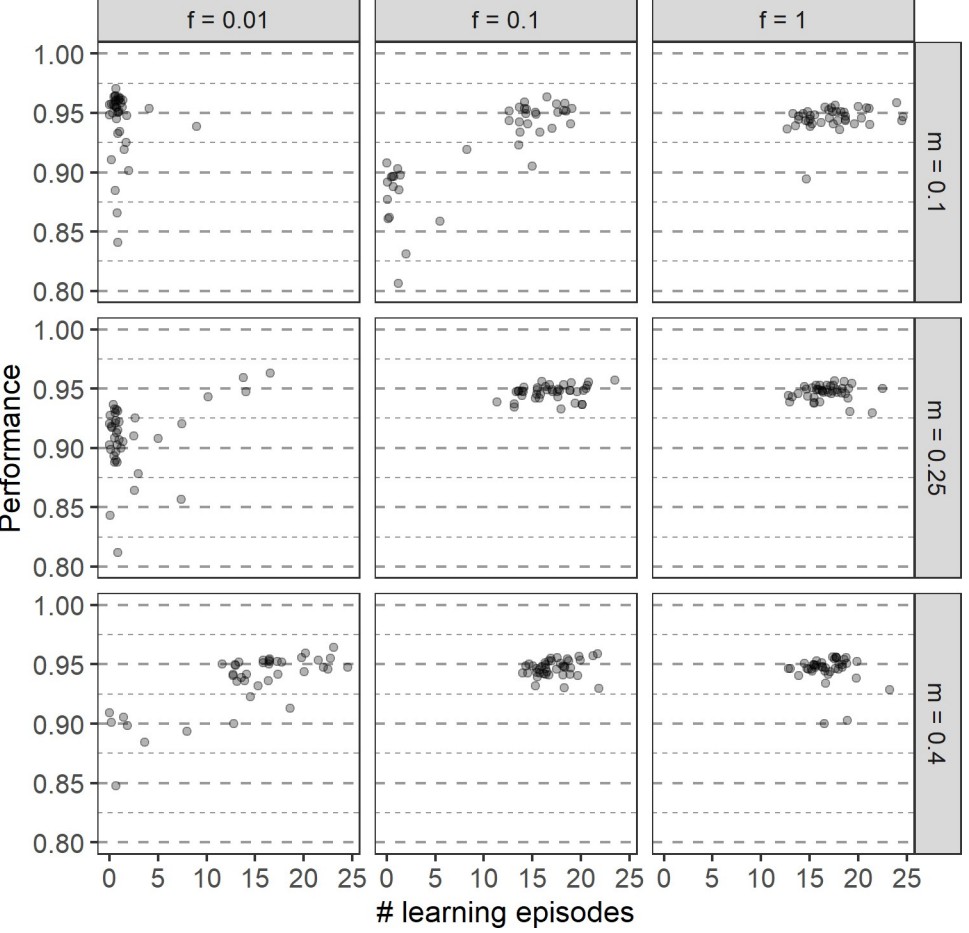

**Fig 8. The relationship between performance and the number of learning episodes in different environmental regimes.** Different columns correspond to different frequencies of environmental change ($f$) and different rows to different magnitudes of environmental change ($m$). Each point represents the average of the population mean over the last 2000 generations of a single replicate. Results of four simulations (all for $f = 0.01$) with low average LE ($< 3$) is not visible, as the average performance was below 0.80. The points are semi-transparent and darker spots indicate that multiple replicates evolved the same values.

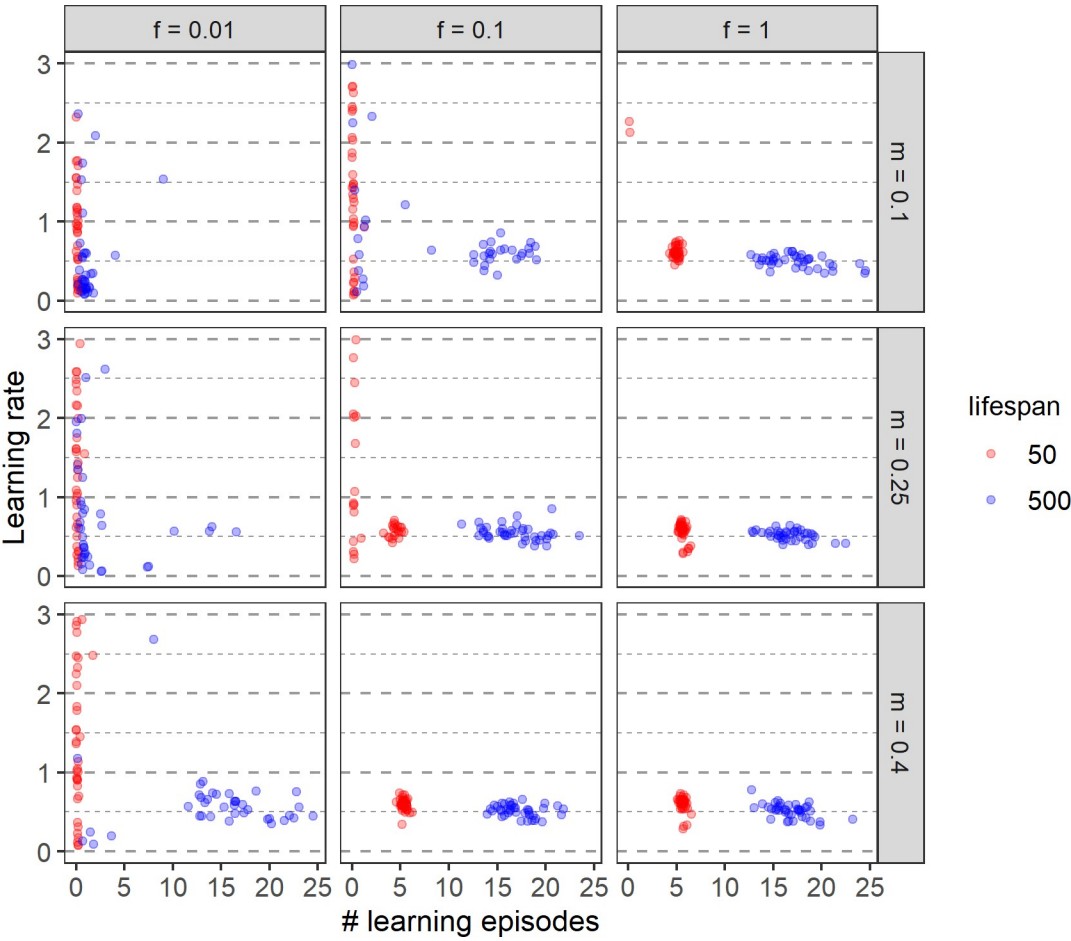

**Fig 9. Effect of lifespan on the evolution of learning.** For two lifespans (50 timesteps: red; 500 timesteps: blue) each panel shows the evolved relationship between the learning rate and the number of learning episodes in 40 replicate simulations. The panels correspond to different environmental regimes: the columns show three frequencies of environmental change (*f*) and the rows three magnitudes of change (*m*). Each point represents the average of the population mean over the last 2000 generations of a single replicate. For clarity, only learning rates up to 3.0 are shown; 36 data points with a learning rate above 3.0, all with a very low number of learning episodes (= no learning), are not visible.

### Effect of lifespan and environment on the evolution of learning

Until now, we considered an organism with a lifetime of 500 timesteps where the exploration-exploitation trade-off is relatively weak because efficient learning can be accomplished within a period of 20 timesteps. In this section, we consider an organism with a lifespan of only 50 timesteps, where the cost of learning is much larger, as each timestep spent learning corresponds to 2% of the organism's lifetime and hence reduces the foraging time considerably.

For various environmental regimes, Fig 9 compares the evolutionary outcome for the two lifespans considered. Not surprisingly, learning is less likely to evolve in the case of a shorter lifespan, especially in those environmental regimes where adaptive tracking is, at least to a certain extent, efficient (a low rate of change and/or a small magnitude of change). When learning evolves despite the short lifespan, the evolved learning period is shorter in absolute terms (smaller number of learning episodes) but longer in relative terms (a larger percentage of the lifespan is spent on learning). As in the case of a long lifespan (Fig 7B and 7C), both the number of learning episodes and the learning rate is practically independent of the magnitude and rate of environmental change. However, the number of learning episodes is less variable,

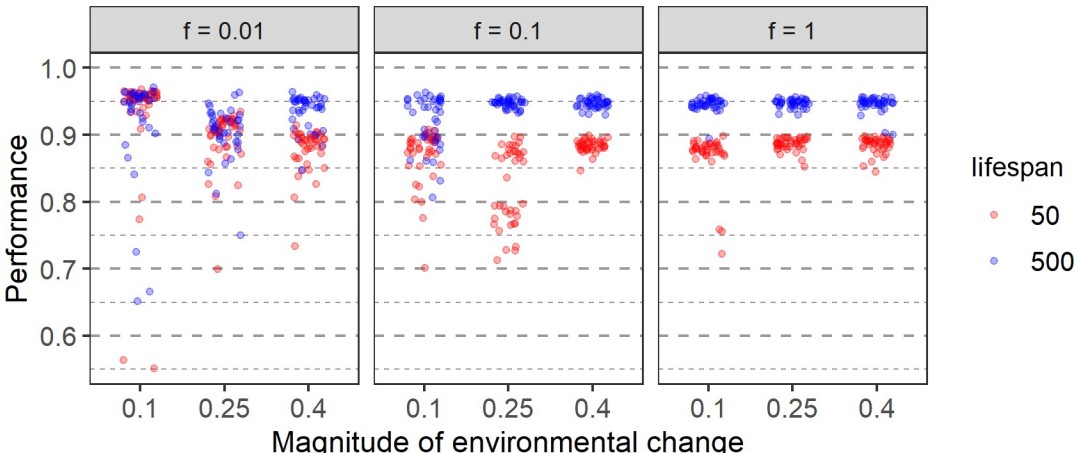

**Fig 10. Effect of lifespan on performance.** Results for two lifespans (50 timesteps: red; 500 timesteps: blue) and nine environmental regimes (defined by the rate *f* and the magnitude *m* of change) are shown. The average population performance in the last 2000 generations of 40 replicate simulations for each parameter set is indicated by coloured dots.

reflecting a stronger selection on the efficient use of every single timestep. Interestingly, when learning evolves in both lifespan conditions, even though the number of learning episodes is clearly different, the evolved learning rate is only slightly higher for a lower lifespan and for some environmental conditions (e.g. for the very frequent environmental change) but is independent of the lifespan for other conditions (see Fig 9 in the main text and Fig E in S1 Appendix).

Fig 10 shows that, whenever learning evolves at all, the resulting performance is smaller in case of a short lifespan. In view of the fact that short-lived organisms spend less time learning, this is not too surprising.

All results presented thus far were based on an environmental quality function with $\sigma = 0.25$ (Fig 2). Here, we consider the implications of a narrower ($\sigma = 0.1$) or a wider ($\sigma = 0.4$) Gaussian function. If the function is narrow, only a small fraction of the available food is of high quality. Therefore, mistakes in choosing among food items potentially lead to a severe loss of fitness. On top of this, environmental change has a more drastic effect. If, for example, the magnitude of environmental change is large (e.g., $m = 0.4$), all high-quality food items in the previous generation become low-quality, while a small range of the previously low-quality items become high-quality after the shift. One would therefore expect a much stronger selection for learning in case of $\sigma = 0.1$. Conversely, selection for learning is expected to be weaker in case of $\sigma = 0.4$.

Fig 11 shows how the evolution of learning (= the evolved duration of the learning period) depends on lifespan and the width of the environmental quality function. Figs E and F in S1 Appendix show the corresponding learning rates and performance levels, respectively. Consider first the case of a narrow quality function ($\sigma = 0.1$). As argued above, one would expect that learning is more important in this case. It is therefore somewhat surprising that learning less easily gets off the ground than in our standard scenario ($\sigma = 0.25$). This may be explained by the fact that efficient learning is difficult to achieve in the case of a narrow quality function. The reason can be that most of the cues sampled during the learning period are of very low (practically zero) quality (cues are sampled randomly); accordingly, these cues provide little information on where the peak of the function is located.

When the quality function is broad ($\sigma = 0.4$) the environmental conditions (frequency and magnitude of change) in which learning evolves are very similar to the ones with $\sigma = 0.25$

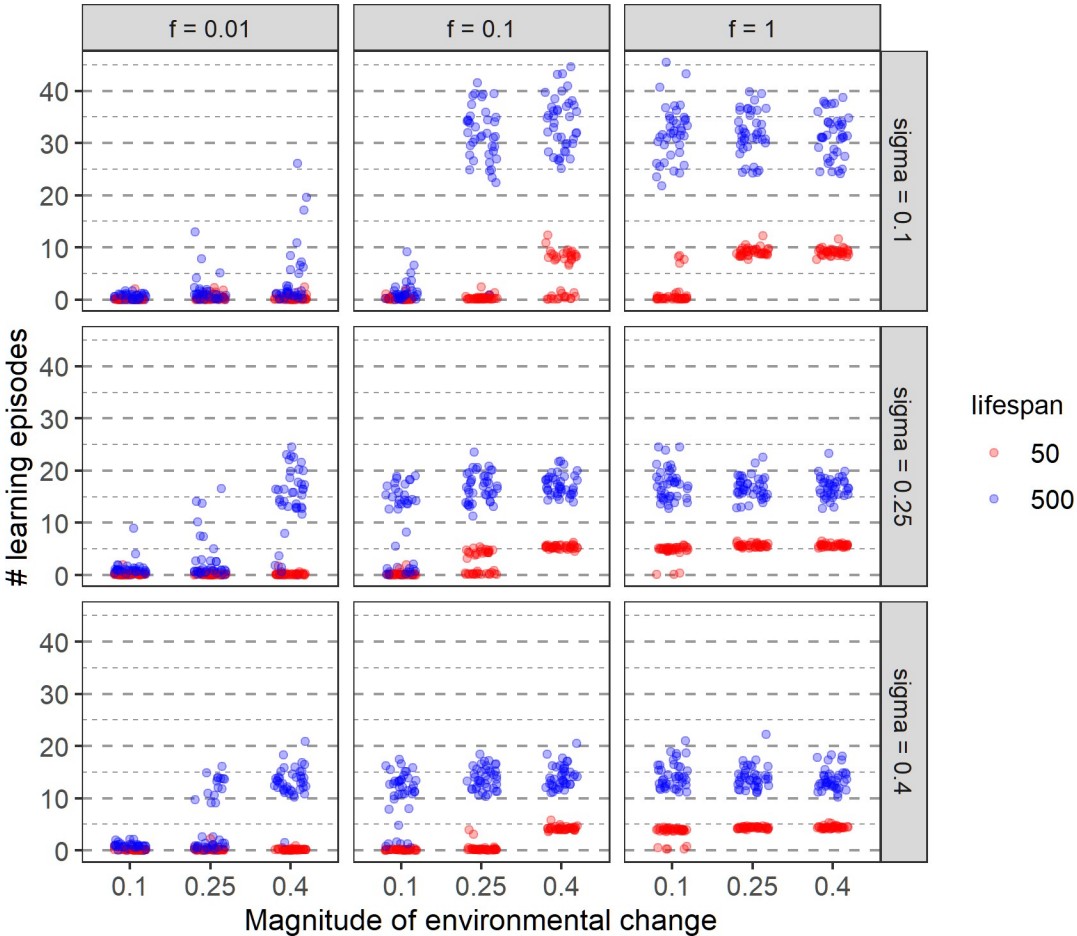

**Fig 11. Effect of lifespan and the width of the quality distribution on the evolution of learning.** Rows correspond to different values of σ (sigma), that is, to different widths of the quality function. Graphical conventions as in the previous figures.

(Fig 11). One clear exception is an environmental change of $m = 0.25$ every 10 generations ($f = 0.1$) for the lifespan of 50. In this case for σ = 0.4 learning never evolves, while it does for roughly half of the replicates for σ = 0.25. It seems that for a relatively small and rare change spending time learning doesn't compensate enough for lost foraging opportunities if food quality changes only slightly around the peak and choosing less optimally does not reduce fitness considerably (for σ = 0.4).

When learning is maintained, the number of learning episodes increases with decreasing σ, while at the same time, performance decreases (Fig F in S1 Appendix) as every mistake is more costly. Similarly, the learning rate decreases with decreasing σ, supporting the general expectation that with a lower number of learning episodes, the learning rate should be larger to allow for faster learning in a shorter time (Fig E in S1 Appendix). However, as noted earlier, within one environmental regime the learning rate might not correlate with the number of learning episodes.

As observed earlier, for a given width of quality distribution, if learning evolves the number of learning episodes and learning rate do not depend on the magnitude and frequency of environmental change (Fig 11 in the main text and Fig E in S1 Appendix).

## Discussion

We presented a novel way of modelling the evolution of learning, using small neural networks and a simple, biology-inspired learning algorithm. We used this model as a tool to answer evolutionary questions that are usually tackled by simple analytical models. Contrary to analytical methods, even a relatively simple network, as studied here, allows for more complex phenotypes and learning strategies and at the same time can lead to efficient adaptation and novel insights into the evolution of learning.

In line with the literature (e.g [7,11,31,38]), the frequency of environmental change had a large effect on the probability that learning will be maintained in the population. The same holds for the magnitude of environmental change, although this aspect of environmental variation has rarely been studied. Usually, environmental change is assumed to be random and there is no correlation between subsequent environmental states (e.g., [11,39] and references therein), even though it is likely not the case in the real world [40]. In our model, when environmental change is infrequent and/or small, genetic tracking evolved. In less stable environments, learning (a "plastic" response) was selected for, but if the change was very frequent and large, individuals made random choices–a strategy resembling bet-hedging. Similar results for the evolution of adaptive tracking, plastic responses, and bet-hedging were observed by Botero *et al*. [37], although they modelled the predictability of the future environment based on the current cues, while in our model different magnitudes and frequencies of environmental change can say something about how much the current environment predicts a future state.

Whenever learning evolves one might expect that the learning strategy (the length of the learning period and the learning rate) is fine-tuned to the environmental change regime. However, surprisingly, we found that the learning strategy was independent of the environmental change regime. To the best of our knowledge, our study is the first to report such a finding. The few studies that investigated such effects reported a relationship between the frequency of environmental change and the evolved or optimal learning parameters [32,33,38,41,42]. We are not aware of any neural network models studying the effect of environmental change regimes on the learning strategy.

The discrepancy between our results and those of previous studies may reflect the higher number of degrees of freedom in our model or the peculiarities of the task itself. In earlier models, learning (governed by a single parameter) affects only one variable., which is sometimes identical to the "phenotype". Under this restriction, different learning parameter values seem necessary to deal with different frequencies of environmental change. In our model, the "phenotype" is more complex (the function ascribing a predicted quality to a range of cues) and learning affects more than one parameter (four network weights). When the environment changes, the relationship between cues and their quality changes. In this case, when efficient learning evolves it lets organisms predict the current location of the environmental peak, independent of when and how much it changed in the past. In the future, it would be interesting to study the evolved learning strategy of networks challenged by a different task, e.g. having to know the quality of all environmental cues, rather than just identifying the ones that are the best in a set of options.

For learning to be efficient, not only a learning strategy must be fine-tuned, but also the underlying neural network. While many networks can perform the task, not all are suitable to learn. For example, our initial networks with random weights did not support efficient learning and the performance of our simulated organisms was initially poor. Randomly initialised neural networks can show a good performance if the learning algorithm is very efficient (e.g. backpropagation) and if the learning consists of many learning steps [43]. However, different studies consistently indicate that networks that evolved initial weights can be trained

significantly faster and better than networks with random initial weights [23,30] (but see the discussion on reservoir computing below). This seems to apply also to our less sophisticated learning mechanism and supports the view that learning and evolution together are more successful than either alone [30,44]. As noted by Mery and Burn "evolution of a combination of learning and innate behavioural responses is probably a common process" [45] but it has very rarely been included in models of the evolution of learning. Neural networks provide an intuitive way of studying the intertwined evolution of innate and learned responses to the environment.

Another interesting finding of our study is that the distribution of resources in the environment strongly affects the probability of learning evolving and the evolved learning strategy. To our knowledge, this has never been observed (or even investigated) before. When only a small fraction of available resources provides nutrients (small σ in our model) then one could expect that learning would be more likely to evolve to allow individuals to find the cues that are linked with profitable food. However, this is not the case and learning is actually less likely to evolve then. It seems that learning is less likely to evolve whenever sampling the environment can lead to frequent encounters with items that do not provide resources and therefore information–"clueless environments" [36]. In this case, spending time learning is relatively more costly, which seems to shift the cost-benefits balance against learning. More studies looking at this aspect of the environment would be welcome.

Not only environmental but also organismal properties can affect the evolution of learning. One obvious one is an organism's lifespan. While it is generally assumed that shorter-lived organisms should invest less in cognitive functions, as the cost of learning is relatively larger while the time to profit from learning is shorter (see the discussion on this topic in [33,46]), there are, to the best of our knowledge, only two modelling studies that investigated the effect of lifespan on learning. Eliassen *et al.* [32] studied learning in the context of the exploration-exploitation trade-off and showed that learners should invest less in learning for shorter expected lifespans (higher external mortality). But the optimal speed of learning depended not only on the expected lifespan but also on the temporal change in the environment. Liedtke & Fromhage [33] built a vastly different and simpler model in which learning reduced the handling time of food items. They showed that the learning speed and the investment in learning (in their model the higher the learning speed the higher costs paid) should be highest for short and intermediate lifespans. Our results are to some extent in line with those of Eliassen *et al.* and the common wisdom that short-lived organisms should invest less in learning. However, in our model, lifespan and the pattern of environmental change have a surprisingly small (or even negligible) effect on the learning speed when learning evolves. Clearly, details of the model assumptions can have a profound effect on the model outcomes. Therefore, additional studies, both theoretical and empirical, on the effect of lifespan on learning are needed.

It is generally accepted that learning is costly. The costs can be manifold, for example, energetic costs of growth and maintenance of the brain tissue, resource allocation trade-offs, or increased mortality due to suboptimal behaviour during the learning phase (see e.g. [7,47–49]). In many studies, the cost of learning is just one of the parameters included *a priori* in the model (see e.g. [11] and references therein). To avoid extra assumptions, the only cost of learning in our model stems from the limited lifespan and the trade-off between exploration (learning) and exploitation (foraging). To this end, our model assumes that an individual's life is divided into two separate stages: learning and foraging. In our model, this simplification is justified by the fact that, by assumption, the environment can only change between generations and is constant within a generation. In such a situation, it is optimal to learn the environmental characteristics as early as possible, in order to profit maximally from learning later in life. In line with this, early life (childhood) is in many animals (including humans) is much more

dedicated to learning than later life [50]. However, most animals can learn during their whole life, and life-long learning is clearly important when the environment changes within a generation. The pattern of change within generations has a great impact on the adaptive value of learning [7,10,32,51], and it may select for specific patterns of learning (so-called "learning schedules"). In principle, our model can cope with within-generation environmental change, but the study of learning schedules would necessitate the inclusion of additional heritable parameters in an already complicated model. Therefore, we do not consider life-long learning here but leave this issue to future studies.

In our model, learning induces a change in the nervous system that usually leads to an improvement in performance. But this is not always the case, e.g. if the learning period is very short. This finding undermines the common assumption that, as long as the learning cues are reliable, learning will always improve performance or even lead to perfect behaviour (e.g. [11,39,52]). Mechanistic approaches like ours are required to elucidate whether and when such an assumption is justified.

Neural network models are especially suited to answer evolutionary questions concerning behaviour as they explicitly incorporate the proximate stimulus-response aspect of behaviour. Optimisation approaches (tend to) neglect the proximate underpinning of adaptations. Neural networks are also suited for more complex problems than the ones that can be tackled with analytical methods [53]. Also, as mentioned earlier, they provide a great opportunity to study the coevolution of innate behaviours together with the evolution of learning mechanisms. In our model, learning affected only part of the network–this approach was inspired by reservoir computing in which also only the weights linked to the output node change, yet the network can still learn complex tasks [26,27,54]. Such a learning mechanism seems very suitable for evolutionary studies as it is unlikely that in early animals a single learning event could affect all connections in the network as is usually assumed in AI applications.

We draw inspiration from reservoir computing in the sense that learning affects only a small subset of network weights. However, our network and the task it needs to perform is much simpler than what is potentially possible with a reservoir computing approach and what biological brains can do [26,27,54]. Promising future projects could incorporate larger networks that not only pass information in one direction (feed-forward) but also include backward connections and feedback loops (allowing for longer-term memory). Such "reservoir" networks could dynamically store various types of inputs, making them available for a diversity of decision-making processes, such as making decisions in the context of foraging, predator avoidance, mating, and social behaviour. Each type of decision (which we henceforth will call a "domain") would be governed by domain-specific output nodes, which are connected to the reservoir. Domain-specific learning could happen as in our model: based on simple mechanisms only affecting the connections to the domain-specific output nodes. Each output node may have its own domain-specific connections with the reservoir, allowing output nodes governing foraging behaviour to tap into different parts of the reservoir than output nodes linked with predator avoidance. By considering mutations that break or create connections from the output nodes to reservoir nodes, this part of the network architecture could evolve by domain-specific selection. Such partial restructuring of the network would likely make behaviour and learning more efficient [27,55], even though learning would affect only a small fraction of the connections. The domain-general reservoir could also be shaped by natural selection. Interestingly, the demands on the reservoir may not be stringent, provided that it is sufficiently complex: as shown in the literature on reservoir computing [26,27,54]. Reservoir-based learning can be very efficient even if the connections between the nodes of the reservoir are quite arbitrary (i.e., if the connection strengths are drawn at random). A network model as sketched above would allow using the same environmental information and network structure to

perform different tasks, as is also seen in the animal brains [27] without making a priori assumptions on what environmental cues are important in different contexts.

In conclusion, we showed that a biologically inspired, yet relatively simple, learning mechanism can evolve to lead to an efficient adaptation in a changing environment. We hope that our model will serve as an inspiration for future work on more challenging research projects and ultimately to a better understanding of the evolution of learning.

## Supporting information

**S1 Appendix. Supplementary figures A-F.** The file contains the following figures: **Fig A. The time course of network performance in a changing environment. Fig B. The effect of a fixed number of learning episodes (LE) on the learning rate evolved in different environmental regimes. Fig C. The relationship between weight strength and the number of learning episodes in different environmental regimes. Fig D. Prediction profiles for evolving networks over time. Fig E. Effect of lifespan and the width of the quality distribution on the evolved learning rate. Fig F. Effect of lifespan and the width of the quality distribution on population performance.**
(PDF)

## Acknowledgments

We thank the MARM and MINDS groups at the University of Groningen for valuable discussions.

## Author Contributions

**Conceptualization:** Magdalena Kozielska, Franz J. Weissing.

**Formal analysis:** Magdalena Kozielska.

**Funding acquisition:** Franz J. Weissing.

**Software:** Magdalena Kozielska.

**Visualization:** Magdalena Kozielska.

**Writing – original draft:** Magdalena Kozielska.

**Writing – review & editing:** Magdalena Kozielska, Franz J. Weissing.

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
