## [Decision Letter · Decision Letter 0]

19 Nov 2023

Dear Kozielska,

Thank you very much for submitting your manuscript "A neural network model for the evolution of learning in changing environments" for consideration at PLOS Computational Biology. As with all papers reviewed by the journal, your manuscript was reviewed by members of the editorial board and by several independent reviewers. The reviewers appreciated the attention to an important topic. Based on the reviews, we are likely to accept this manuscript for publication, providing that you modify the manuscript according to the review recommendations.

Please take the comments of the three reviewers into consider (in particular the issues pointed out by Reviewer 1).

Sincerely,

Xingru Chen, Ph.D.

Guest Editor

PLOS Computational Biology

Zhaolei Zhang

Section Editor

PLOS Computational Biology

Please take the comments of the three reviewers into consider (in particular the issues pointed out by Reviewer 1).

Reviewer's Responses to Questions

**Comments to the Authors:**

Reviewer #1: Generally speaking I am really enthusiastic about the work presented in this MS. As the authors point out there are relatively few formal models of the evolution of learning and none that explicitly consider how network-based learning is likely to evolve. In addition, the analyses are well executed and clearly discussed, and so the work will almost certainly have an impact on the field. I have only one main issue I feel needs to be explicitly considered in the MS to maximise its potential.

Most of the functional ("behavioural gambit") logic underpinning existing understanding about the selective advantages of learning emphasise the key role of within generation (within life time) patterns of variation with the received wisdom being that learning is most strongly selected for when within-generation variation is substantial, varies between generations and potentially trackable. However, the authors only consider ecological change happening between generations here. That learning is still favoured at all seems to me to be because there is a noisy cue that can be at least partially resolved via repeated sampling. Now none of this renders the analysis uninteresting - the selective value of being able to resolve such cue error via learning is not something that has been widely considered in the evolutionary literature to my knowledge. Nevertheless, i do feel that the authors need to acknowledge this issue and discuss how their formulation relates to such "classic" ideas about how ecological change affects the evolutionary value of learning. This received wisdom is derived from Dave Stephens' early discussions and analyses culminating in his seminal model in Stephens (1991), which I note the authors do not cite.

Stephens, D. W. (1991). Chnage, regularity, and value in the evolution of animal learning. Behavioral Ecology, 2(1), 77–89. https://doi.org/10.1093/beheco/2.1.77

Reviewer #2: See attached review report

Reviewer #3: This study introduces a novel approach to modeling learning using small neural networks and a biologically inspired learning algorithm that selectively impacts part of the network based on the disparity between expectations and reality. This model offers insights into how natural selection shapes learning, and authors applied it to explore the evolution of learning under diverse environmental conditions, considering different trade-offs between exploration (learning) and exploitation (foraging).

In the individual-based simulations, efficient learning regularly evolved. However, consistent environments with minimal change and short-lived organisms, unable to dedicate much time to exploration, posed challenges to the evolution of learning, consistent with previous findings. Once learning evolved, surprisingly, the characteristics of the learning strategy (learning period duration and learning rate) and average performance after learning were minimally influenced by the frequency and magnitude of environmental changes. Conversely, an organism's lifespan and the resource distribution in the environment significantly impacted the evolved learning strategy.

Notably, a longer learning period did not universally lead to improved performance, suggesting variability in the effectiveness of evolved neural networks. Testing results demonstrate that a relatively simple, biologically inspired learning mechanism can evolve to facilitate efficient adaptation in dynamic environments.

**Have the authors made all data and (if applicable) computational code underlying the findings in their manuscript fully available?**

Reviewer #1: Yes

Reviewer #2: Yes

Reviewer #3: Yes

PLOS authors have the option to publish the peer review history of their article (what does this mean?). If published, this will include your full peer review and any attached files.

Reviewer #1: No

Reviewer #2: **Yes: **Xin Wang

Reviewer #3: No

Figure Files:

Data Requirements:

Reproducibility:

References:

---

## [Decision Letter · Decision Letter 1]

18 Jan 2024

Dear Kozielska,

We are pleased to inform you that your manuscript 'A neural network model for the evolution of learning in changing environments' has been provisionally accepted for publication in PLOS Computational Biology.

Best regards,

Xingru Chen, Ph.D.

Guest Editor

PLOS Computational Biology

Zhaolei Zhang

Section Editor

PLOS Computational Biology

Reviewer's Responses to Questions

**Comments to the Authors:**

Reviewer #1: I am happy with the authors' response to my comments and the changes to the MS they made. I look forward to seeing this in print.

Reviewer #2: I appreciate the authors' great efforts on addressing all my concerns and I believe their work will arise much attention in the field. Therefore I'm happy to recommend accepting it for publication in PLOS Computational Biology.

Reviewer #4: This study introduces a novel approach to modeling learning using small neural networks and a simple, biology-inspired learning algorithm, offering insights into the evolution of this crucial adaptation mechanism. Unlike many existing models that focus on simple phenotypes or make biologically unrealistic assumptions, the approach incorporates the complexities of neural networks and considers evolutionary questions.

In the simulations, this paper explore the evolution of learning under diverse environmental conditions, examining different trade-offs between exploration (learning) and exploitation (foraging). The results show that efficient learning readily evolves in the individual-based simulations. However, consistent environments or short-lived organisms pose challenges to the evolution of learning, consistent with previous findings.

Surprisingly, once learning evolves, the characteristics of the learning strategy (e.g., learning period duration and learning rate) and average performance after learning are minimally affected by the frequency and magnitude of environmental changes. Instead, an organism's lifespan and the resource distribution in the environment significantly impact the evolved learning strategy. Shorter lifespans and narrow resource distributions reduce the likelihood of learning evolution.

Notably, a longer learning period does not universally lead to better performance, suggesting variability in the effectiveness of evolved neural networks. Overall, this study demonstrates that a biologically inspired, relatively simple learning mechanism can evolve to facilitate efficient adaptation in dynamic environments.

Authors also well figured out the questions and concerns raised in previous reviews; thus, suggest to accept.

**Have the authors made all data and (if applicable) computational code underlying the findings in their manuscript fully available?**

Reviewer #1: Yes

Reviewer #2: Yes

Reviewer #4: Yes

PLOS authors have the option to publish the peer review history of their article (what does this mean?). If published, this will include your full peer review and any attached files.

Reviewer #1: No

Reviewer #2: **Yes: **Xin Wang

Reviewer #4: No

---

## [Editor Report · Acceptance letter]

25 Jan 2024

PCOMPBIOL-D-23-01364R1 

A neural network model for the evolution of learning in changing environments

Dear Dr Kozielska,

I am pleased to inform you that your manuscript has been formally accepted for publication in PLOS Computational Biology. Your manuscript is now with our production department and you will be notified of the publication date in due course.

With kind regards,

Zsofia Freund
